# ASTUTE RAG: OVERCOMING IMPERFECT RETRIEVAL AUGMENTATION AND KNOWLEDGE CONFLICTS FOR LARGE LANGUAGE MODELS

## ABSTRACT

Retrieval-Augmented Generation (RAG), while effective in integrating external knowledge to address the limitations of large language models (LLMs), can be undermined by *imperfect* retrieval, which may introduce irrelevant, misleading, or even malicious information. Despite its importance, previous studies have rarely explored the behavior of RAG through joint analysis on how errors from imperfect retrieval attribute and propagate, and how potential conflicts arise between the LLMs' internal knowledge and external sources. We find that imperfect retrieval augmentation might be inevitable and quite harmful, through controlled analysis under realistic conditions. We identify the *knowledge conflicts* between LLM-internal and external knowledge from retrieval as a bottleneck to overcome in the post-retrieval stage of RAG. To render LLMs resilient to imperfect retrieval, we propose ASTUTE RAG, a novel RAG approach that *adaptively* elicits essential information from LLMs' internal knowledge, *iteratively* consolidates internal and external knowledge with *source-awareness*, and finalizes the answer according to information reliability. Our experiments using Gemini and Claude demonstrate that ASTUTE RAG significantly outperforms previous robustness-enhanced RAG methods. Notably, ASTUTE RAG is the only approach that matches or exceeds the performance of LLMs without RAG under worst-case scenarios. Further analysis reveals that ASTUTE RAG effectively resolves knowledge conflicts, improving the reliability and trustworthiness of RAG systems.

## 1 INTRODUCTION

Retrieval augmented generation (RAG) has become the standard approach for large language models (LLMs) to tackle knowledge-intensive tasks (Guu et al., 2020; Lewis et al., 2020). Prior works mainly leverage RAG to address the inherent knowledge limitations of LLMs, effectively integrating missing information and grounding to reliable sources. However, recent research has highlighted a significant drawback that RAG might rely on *imperfect retrieval* results, including irrelevant, misleading, or even malicious information, which eventually leads to inaccurate LLM responses (Chen et al., 2024a; Xiang et al., 2024; Zou et al., 2024). For example, when asked about the practice of eating rocks, LLMs might cite misleading information, such as a satirical news source claiming that one should consume at least one rock per day.[1] The occurrence of imperfect retrieval augmentation is inevitable, driven by factors such as corpus quality limitations (Shao et al., 2024), the reliability of retrievers (Dai et al., 2024), and the complexity of the queries (Su et al., 2024). This poses a significant challenge to the trustworthiness of RAG.

While there have been some pioneering analyses of RAG on noisy context (Chen et al., 2024a; Zou et al., 2024; Xiang et al., 2024), a more comprehensive analysis and solution is needed to explore the propagation of realistic errors in retrieval results, leading to *knowledge conflicts* (Longpre et al., 2021) between LLMs and context, and ultimately, RAG failures. To this end, we conduct comprehensive analyses on the occurrence of imperfect retrieval augmentation and its impact on LLM behavior under realistic conditions (Section 2). We conduct controlled experiments on a diverse range of general, domain-specific, and long-tail questions from NQ (Kwiatkowski et al., 2019), TriviaQA

---

[1] https://www.bbc.com/news/articles/cd11gzejgz4o.

Figure 1: Knowledge conflicts between the LLMs' internal knowledge and retrieved knowledge from external sources. We report the overall results with Claude under the setting in Section 4.1.

(Joshi et al., 2017), BioASQ (Tsatsaronis et al., 2015), and PopQA (Mallen et al., 2023). We observe that imperfect retrieval augmentation is widespread even with adept real-world search engine (such as Google Search with Web as corpus) – roughly 70% retrieved passages do not directly contain true answers, leading to the impeded performance of LLM with RAG augmentation.

These findings underscore the potential severity of the imperfect retrieval issue in real-world RAG and highlight the widespread existence of knowledge conflicts as the bottleneck to overcome it. Recent studies demonstrate that LLM-internal and external knowledge offer distinct advantages, but LLMs often struggle to consolidate conflicting information reliably, failing to respond based on collective knowledge (Mallen et al., 2023; Tan et al., 2024; Xie et al., 2024; Jin et al., 2024). This raises the following research question: *Is there an effective method to combine internal (from LLMs' pretrained weights) and external (from specific corpora or knowledge bases) knowledge for more reliable RAG?* Previous work has widely explored using external knowledge to enhance LLMs through RAG. We seek to further leverage LLMs' internal knowledge to recover from RAG failures

Motivated by these important real-world challenges, we propose ASTUTE RAG (Section 3), a novel RAG approach designed to be resilient to imperfect retrieval augmentation, while preserving RAG grounding effect when RAG is reliable. To this end, ASTUTE RAG needs effectively differentiate the reliability of the LLM's intrinsic knowledge and the external information retrieved in RAG, utilizing each only when trustworthy and ensuring proper integration. Specifically, ASTUTE RAG initially elicits information from LLMs' internal knowledge to explicitly complement the passages retrieved from external sources. Then, ASTUTE RAG conducts source-aware knowledge consolidation of information from various internal and external sources. The desiderata is combining consistent information, identifying conflicting information, and filtering out irrelevant information. Finally, ASTUTE RAG proposes answers based on each group of consistent passages and compares the answers from different passage groups to determine the final answer. Our experiments involving Gemini and Claude[2] on various datasets (Section 4) demonstrate the superior performance of ASTUTE RAG compared to previous RAG approaches designed to be robust against retrieval corruptions. Moreover, ASTUTE RAG consistently outperforms baselines across different retrieval quality levels. Notably, ASTUTE RAG is the only RAG method that achieves performance comparable to or even surpassing conventional use of LLMs under the worst-case scenario where all retrieved passages are unhelpful. Further analysis reveals the effectiveness of ASTUTE RAG in resolving knowledge conflicts between internal and external knowledge.

To conclude, our core contributions are threefold. First, we analyze RAG under realistic conditions, identifying imperfect retrieval augmentation as a significant contributor to RAG failures and pinpointing knowledge conflicts as the primary bottleneck in overcoming it. Second, we propose ASTUTE RAG, which explicitly addresses conflicts between LLM-internal and external knowledge, thereby recovering from RAG failures. Third, experiments with various LLMs and datasets demonstrate the effectiveness of ASTUTE RAG, even in the most challenging scenarios.

## 2 IMPERFECT RETRIEVAL: THE PITFALL OF RAG

To better showcase the common real-world challenges and to make better motivate for improved methodological designs, we evaluate retrieval quality, end-to-end RAG performance, and knowledge

---

[2]https://www.anthropic.com/claude

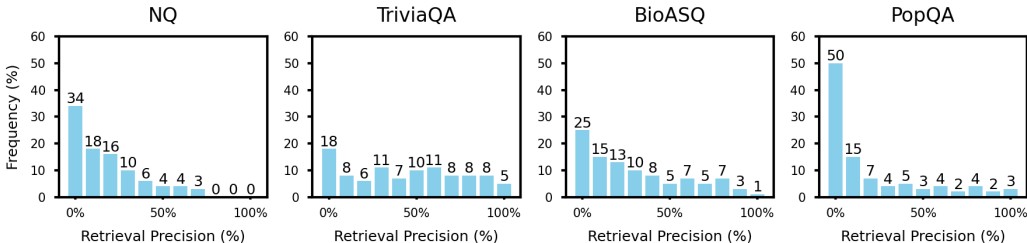

Figure 2: Imperfect retrieval (samples with low retrieval precision) is prevalent in real-world RAG.

conflicts on a controlled set of data. The selected data encompass a diverse range of general, domain-specific, and long-tail questions from NQ (Kwiatkowski et al., 2019), TriviaQA (Joshi et al., 2017), BioASQ (Tsatsaronis et al., 2015), and PopQA (Mallen et al., 2023). Our analysis is based on *realistic* retrieval results with Google Search[3] as the retriever and the Web as the corpus. This setting allows us to analyze the severity of imperfect retrieval in real-world RAG. Overall, we sample 1K short-form QA instances from these datasets, and pair each instance with 10 retrieved passages.

**Imperfect retrieval is common.** We examine the occurrence of correct answers in retrieved passages as an approximation of retrieval quality. Since we mainly focus on short-form QA which provides most variants of the correct answer for each question, the approximation through string matching can give us a rouge intuition of how precise the retrieval result is. Specifically, we define the retrieval precision as the ratio of passages containing the correct answer for each instance:

$$\text{Retrieval Precision} = \frac{\{\text{number of retrieved passages containing correct answer}\}}{\{\text{number of total retrieved passages}\}}$$

As shown in Figure 2, although instances from different datasets exhibit different data distributions, imperfect retrieval is prevalent. Specifically, $\sim$20% of the overall data have no mentions of the correct answer within any retrieved passage, including 34% on NQ, 18% on TriviaQA, 24% on BioASQ, and 50% on PopQA. This finding also aligns with previous observation on information retrieval (Thakur et al., 2024), that highlights that the number of positive passages can be very limited.

**Imperfect retrieval leads to RAG failures.** We further analyze the relation between retrieval quality and RAG performance. We compare the performance of Claude 3.5 Sonnet, with and without RAG and report the results by retrieval precision in Figure 5. In general, RAG is helpful when the retrieval precision is not lower than 20%. When the retrieval precision is close to 0, the model with RAG performs much worse than without RAG, indicating that imperfect retrieval augmentation can be the cause of RAG failures. This finding aligns with the previous observation from Yu et al. (2024) that adding more retrieved passages does not necessarily lead to better performance, as the additional passages might reduce the retrieval precision.

**Knowledge conflicts widely exist in RAG failures.** We provide an in-depth analyses of knowledge conflicts between LLMs' internal knowledge and retrieved passages from external sources. With Claude 3.5 Sonnet as the LLM, Figure 1 shows that 19.2% of the overall data exhibit knowledge conflicts, where either the answer with or without RAG is correct. Among the conflicting cases, the internal knowledge is correct on 47.4% of them, while the external knowledge is correct on the remaining 52.6%. These results emphasize the importance of *effectively combining the internal and external knowledge to overcome the inherent limitation of relying solely on either source*. However, previous work (Tan et al., 2024; Xie et al., 2024; Jin et al., 2024) show that LLMs might respond based on misleading information rather than comprehensive understanding of the conflicting knowledge in this context.

## 3 ASTUTE RAG: OVERCOMING THE PITFALL

We begin with formulating the problem of imperfect retrieval in RAG (Section 3.1). We then provide an overview of ASTUTE RAG, designed to overcome this problem (Section 3.2). Subsequently, we delve into the three major steps of ASTUTE RAG, including adaptive generation of internal knowledge (Section 3.3), source-aware knowledge consolidation (Section 3.4), and answer finalization (Section 3.5).

---

[3]https://developers.google.com/custom-search/v1/overview

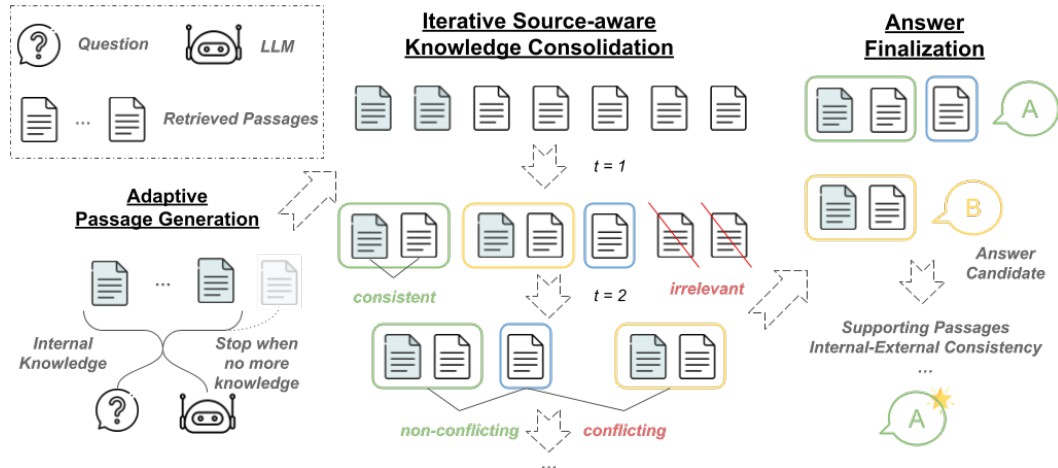

Figure 3: Overview of the proposed ASTUTE RAG framework. ASTUTE RAG is designed to better combine the information from the external sources (e.g. web, domain-specific corpora, knowledge bases) and internal knowledge of the LLMs by employing a consolidation mechanism to address the information conflicts, which eventually leads to better quality generated outputs.

## 3.1 PROBLEM FORMULATION

Our objective is to mitigate the effects of imperfect retrieval augmentation, resolve knowledge conflicts between the LLM's internal knowledge and external sources (such as custom/public corpora and knowledge bases), and ultimately produce more accurate and reliable responses from LLMs.

Given a set of retrieved passages from external sources $E = [e_1, \ldots, e_n]$, a pre-trained LLM $\mathcal{M}$ (accessible through prediction-only APIs, encompassing commercial black-box ones), and a query $q$, the task is to generate the corresponding correct answer $a^*$. Notably, this setting is orthogonal to prior work on improving the retriever, training LLMs, or conducting adaptive retrieval, which are mainly preliminary steps.

## 3.2 OVERVIEW OF THE FRAMEWORK

ASTUTE RAG is designed to better leverage collective knowledge from both internal knowledge of LLMs and external corpus, for more reliable responses. As shown in Figure 3 and Algorithm 1, ASTUTE RAG starts from acquiring the most accurate, relevant, and thorough passage set from the LLMs' internal knowledge. Then, internal and external knowledge are consolidated in an iterative way, by comparing the generated and retrieved passages. Finally, the reliability of conflicting information is compared and the final output is generated according to the most reliable knowledge.

## 3.3 ADAPTIVE GENERATION OF INTERNAL KNOWLEDGE

In the first step, we elicit internal knowledge from LLMs. This LLM-internal knowledge, reflecting the consensus from extensive pre-training and instruction-tuning data, can supplement any missing information from the limited set of retrieved passages and enable mutual confirmation between LLM-internal and external knowledge. This is especially valuable when the majority of retrieved passages might be irrelevant or misleading. Specifically, we prompt LLMs to generate passages based on the given question $q$, following Yu et al. (2023a). While Yu et al. (2023a) primarily focused on generating diverse internal passages, we emphasize the importance of reliability and trustworthiness of generated passages. To achieve this goal, we enhance the original method with *constitutional principles* and *adaptive generation*.

Inspired by Constitutional AI (Bai et al., 2022), we provide **constitutional principles** indicating the desired properties of internal passages in the prompt $p_{gen}$ (see Appendix A for details) to guide their generation, emphasizing that the generated passages should be accurate, relevant, and hallucination-free. Moreover, we allow the LLM to perform **adaptive generation** of passages in its internal knowledge. The LLM can decide how many passages to generate by itself. Rather generating a fix number of passages, we request the LLM to generate at most $\hat{m}$ passages, each covering distinct

---

**Algorithm 1** ASTUTE RAG

---

**Require:** Query $q$, Retrieved Passages $E = [e_1, \ldots, e_n]$, Large Language Model $\mathcal{M}$, Number of
    Iteration $t$, Max Number of Generated Passages $\hat{m}$, Prompt Templates $p_{gen}, p_{con}, p_{ans}$
 1: Adaptively generate passages: $I \leftarrow \mathcal{M}(p_{gen}, q, \hat{m})$          ▷ Section 3.3
 2: Combine internal and external passages: $D_0 \leftarrow E \oplus I$
 3: Assign passage sources: $S_0 \leftarrow [\mathbb{1}_{\{d \in E\}} \text{for } d \text{ in } D_0]$
 4: **if** $t > 1$ **then**
 5:     **for** $j = 1, \ldots, t-1$ **do**                ▷ Section 3.4
 6:         Consolidate knowledge: $\langle D_{j+1}, S_{j+1} \rangle \leftarrow \mathcal{M}(p_{con}, q, \langle D_0, S_0 \rangle, \langle D_j, S_j \rangle)$
 7:     **end for**
 8:     Finally consolidate and answer: $a \leftarrow \mathcal{M}(p_{ans}, q, \langle D_0, S_0 \rangle, \langle D_{t-1}, S_{t-1} \rangle)$    ▷ Section 3.5
 9: **else**
10:     Consolidate knowledge and finalize the answer: $a \leftarrow \mathcal{M}(p_{ans}, q, \langle D_0, S_0 \rangle)$
11: **end if**
12: **return** $a$

---

information, and to directly indicate if no more reliable information is available. This adaptive approach allows the LLM to generate fewer passages (or even no passages at all) when the useful information within internal knowledge is limited and more passages when there are multiple feasible answers in the internal knowledge. In this step, the LLM generates $m \le \hat{m}$ passages based on its internal knowledge:

$$I = [i_1, \ldots i_m] = \mathcal{M}(p_{gen}, q, \hat{m}).$$

### 3.4 ITERATIVE SOURCE-AWARE KNOWLEDGE CONSOLIDATION

In the second step, we employ the LLM to explicitly consolidate information from both passages generated from its internal knowledge and passages retrieved from external sources. Initially, we combine passages from both internal and external knowledge sources $D_0 = E \oplus I$.

We additionally ensure **source-awareness** by providing the source of each passage to LLMs when consolidating knowledge. The source information (internal or external, such as a website) is helpful in assessing the reliability of passages. Here, we provide the passage source as $S_0 = [\mathbb{1}_{\{d \in E\}} \text{for } d \text{ in } D_0]$.

To consolidate knowledge, we prompt the LLM (with $p_{con}$ in Appendix A) to identify consistent information across passages, detect conflicting information between each group of consistent passages, and filter out irrelevant information. This step would regroup the unreliable knowledge in input passages into fewer refined passages. The regrouped passages will also attribute their source to the corresponding one or more input passages

$$\langle D_{j+1}, S_{j+1} \rangle = \mathcal{M}(p_{con}, q, \langle D_0, S_0 \rangle, \langle D_j, S_j \rangle).$$

We find that this is especially helpful in comparing the reliability of conflicting knowledge and addressing knowledge conflicts. Moreover, this knowledge consolidation process can run **iteratively** for $t$ times to improve the context to be more and more useful. Users can assign a larger number of iterations when the context is lengthy.

### 3.5 ANSWER FINALIZATION

In the last step, we prompt the LLM (with $p_{ans}$ in Appendix A) to generate one answer based on each group of passages ($\langle D_t, S_t \rangle$), and then compare their reliability and select the most reliable one as the final answer. This comparison allows the LLM to comprehensively consider knowledge source, cross-source confirmation, frequency, and information thoroughness when making the final decision. Notably, this step can be merged into the last knowledge consolidation step to reduce the inference complexity (the amount of prediction API calls) using a combined prompt:

$$a = \mathcal{M}(p_{ans}, q, \langle D_0, S_0 \rangle, \langle D_t, S_t \rangle).$$

When $t = 1$, the initial passages will be input to the model directly for knowledge consolidation and subsequent answering: $a = \mathcal{M}(p_{ans}, q, \langle D_0, S_0 \rangle)$.

## 4 EXPERIMENTS

We evaluate the effectiveness of ASTUTE RAG on overcoming imperfect retrieval augmentation and addressing knowledge conflicts. In this section, we first introduce the experiment setting in detail (Section 4.1). Then, we compare the performance of ASTUTE RAG with various baselines on diverse datasets (Section 4.2). Finally, we provide in-depth analyses (Section 4.3).

### 4.1 EXPERIMENTAL SETTINGS

**Datasets and metrics.** We conduct experiments on the data collected in Section 2 consisting of data from NQ, TriviaQA, BioASQ, and PopQA. For each instance from these datasets, we provide 10 passages collected under a realistic retrieval setting: for each question in our benchmark, we query Google Search to retrieve the top 30 results and select the first 10 accessible websites. From each retrieved website, we extract the paragraph corresponding to the snippet provided in Google Search results as the retrieved passage.. Most of the retrieval results contains natural noise with irrelevant or misleading information. We do not consider enhancements to the retrieval side, such as query rewriting, as such enhancements are typically already incorporated into commercial information retrieval systems. Notably, we do not select questions or annotate answers based on the retrieval results. This setting allows us to analyze the severity of imperfect retrieval in real-world RAG. It distinguishes our benchmark from previous ones that employ synthetic retrieval corruptions or that unintentionally reduce the frequency of imperfect retrieval with biased construction protocols (Chen et al., 2024a; Yang et al., 2024). We also evaluate our method on RGB (Chen et al., 2024a), a RAG diagnostic benchmark evaluating several crucial RAG abilities. Specifically, we choose the English subset of RGB focusing on noise robustness. The benchmark have positive and negative passage sets for each question. We select five negative documents per question as the context to form a worst-case scenario. All the data in these datasets are short-form QA. Following previous work (Xiang et al., 2024; Wei et al., 2024; Mallen et al., 2023), a model response is considered correct if it contains the ground-truth answer. To enhance evaluation reliability, we prompt LLMs to enclose the exact answer within special tokens, extracting them as the final responses.

**General Settings of LLMs and RAG.** We conduct experiments on both close-source and open-source LLMs of different scales, including Gemini 1.5 Pro[4] (gemini-1.5-pro-002), Claude 3.5 Sonnet[5] (claude-3-5-sonnet@20240620), Mistral-Large (128B;version 2407), and Mistral-Nemo (12B; version 2407). The generation temperature is set to 0 and the maximum output tokens is set to 1,024, if not specified otherwise. By default, the passages are presented in the prompt by reversed order. All experiments are under the zero-shot setting for controlled evaluation, where no demonstrations for QA or method-specific steps are provided.

**Baselines.** We compare ASTUTE RAG with various RAG methods designed for enhanced robustness and representative inference strategies designed to improve response trustworthiness. *USC* (Chen et al., 2024b) is the universal self-consistency method that samples multiple LLM responses given the same context and aggregates the answers. It provides a reference of naive improvements using additional API calls. The temperature for sampling responses in this baseline is set to 0.7. *Genread* (Yu et al., 2023a) augments retrieved passages with LLM-generated passages. It provide a reference of presenting passages from both internal and external knowledge in the prompt without effectively combining them. *RobustRAG* (Xiang et al., 2024) aggregates answers from each independent passage to provide certifiable robustness. We use the keyword aggregation variant as it is shown to be the best-performing variant on advanced LLMs. *InstructRAG* (Wei et al., 2024) instructs the LLM to provide a rationale connecting the answer with information in passages. For a fair comparison, we use the instructions without training or in-context learning. *Self-Route* (Xu et al., 2024) adaptively switches between LLMs with and without RAG.[6] This baseline provides a reference of switching between LLMs' internal and external knowledge.

**Implementation Details of ASTUTE RAG.** The prompt templates for ASTUTE RAG can be found in Appendix A. By default, we use 2 API calls per query, setting $t = 1$ to merge the prompt for

---

[4]https://deepmind.google/technologies/gemini/pro/

[5]https://www.anthropic.com/news/claude-3-5-sonnet

[6]The original Self-Route switches between RAG and long-context LLMs, while our implementation switches between LLMs with and without RAG according to the problem formulation in this paper.

| Method | #API Calls | NQ | TriviaQA | BioASQ | PopQA | Overall |
|---|---|---|---|---|---|---|
| *Claude 3.5 Sonnet (20240620)* | | | | | | |
| No RAG | 1 | 47.12 | 81.98 | 50.35 | 29.78 | 54.51 |
| RAG | 1 | 44.41 | 76.68 | 58.04 | 35.96 | 55.47 |
| USC (Chen et al., 2024b) | 4 | 48.14 | 80.21 | 61.54 | 37.64 | 58.73 |
| GenRead (Yu et al., 2023a) | 2 | 42.03 | 74.20 | 56.99 | 34.27 | 53.55 |
| RobustRAG (Xiang et al., 2024) | 11 | 47.80 | 78.09 | 56.29 | 37.08 | 56.53 |
| InstructRAG (Wei et al., 2024) | 1 | 47.12 | 83.04 | 58.04 | 41.01 | 58.83 |
| Self-Route (Xu et al., 2024) | 1-2 | 47.46 | 78.80 | 59.09 | 41.01 | 58.06 |
| ASTUTE RAG (t=1) | 2 | 52.20 | 84.10 | 60.14 | 44.38 | 61.71 |
| ASTUTE RAG (t=2) | 3 | 53.22 | **84.45** | 61.89 | **44.94** | 62.67 |
| ASTUTE RAG (t=3) | 4 | **53.56** | **84.45** | **62.24** | **44.94** | **62.86** |

Table 1: Main results on Claude under zero-shot setting, showing the accuracy of different benchmark methods vs. ASTUTE RAG, along with their prediction complexity, in number of prediction API calls. Best scores are in bold.

knowledge consolidation and answer finalization. For adaptive generation of internal knowledge, we prompt the LLM to generate no more than one passage.

## 4.2 MAIN RESULTS

**Performance on RAG under realistic retrieval.** Table 1 and Table 3 presents the results on data with realistic retrieval augmentation for each dataset. By comparing RAG and No RAG, we find that retrieved passages might not always bring benefits to downstream performance – on NQ and TriviaQA, RAG performance lags behind No RAG. We attribute this to that the questions being covered by the LLM's internal knowledge and the noise in retrieval results misleading the LLM. In contrast, on BioASQ and PopQA, which focus on domain-specific and long-tail questions, RAG significantly improves LLM performance. However, due to imperfect retrieval augmentation, the absolute performance still remains to be unsatisfactory. Among all baselines, no single method consistently outperforms others across all datasets. This observation highlights that these baselines are tailored to distinct settings and may not be universally applicable. For instance, InstructRAG is more effective on TriviaQA, achieving the best performance among all baselines with both Claude and Gemini. In contrast, Self-Route performs better than InstructRAG on both NQ and BioASQ. Moreover, RobustRAG achieves very different performance when applied to Gemini and Claude. Through in-depth analysis, we find that RobustRAG with Gemini exhibits a high refusal rate (refuse to answer) in responses. We attribute this instability to the varying method designs of the baselines, which are tailored for different scenarios, resulting in inconsistent improvement across datasets. Overall, InstructRAG and Self-Route demonstrates the best performance among all baselines when applied to Claude and Gemini respectively. We also note that increasing the number of API calls does not necessarily correlate with improved performance. The results remain consistent across Mistral-Large and Mistral-Nemo, as shown in Table 4.

ASTUTE RAG consistently outperforms baselines across all datasets of different properties. The overall improvement compared with the best baseline is relatively 6.85% on Claude and 4.13% on Gemini, and the improvements in domain-specific questions are much higher. These results highlight the effectiveness of ASTUTE RAG in overcoming imperfect retrieval augmentation. On Claude, adding more iteration of knowledge consolidation leads to consist improvement. The improvement margin becomes lower when

| Method | EM |
|---|---|
| RAG | 32.97 |
| Instruct RAG | 34.99 |
| Self-Route | 34.47 |
| Astute RAG | 36.81 |

Table 2: Performance on ASQA.

$t$ becomes larger. This is because after each iteration, the remaining improvement space for knowledge consolidation becomes smaller. On Gemini, increasing $t$ primarily benefits BioASQ and PopQA. These two datasets rely more heavily on external knowledge, and iterative knowledge consolidation helps mitigate noise within this external information. Performance on NQ and TriviaQA does not improve further when $t$ reaches 3. We attribute this to the less critical role of external knowledge in these datasets. For setting consistency and efficiency, we set the parameter $\hat{m}$ to a smaller value, limiting the influence of internal knowledge.

| Method | #API Calls | NQ | TriviaQA | BioASQ | PopQA | Overall |
|---|---|---|---|---|---|---|
| *Gemini 1.5 Pro (002)* | | | | | | |
| No RAG | 1 | 44.75 | 80.21 | 45.80 | 25.28 | 51.34 |
| RAG | 1 | 42.71 | 75.97 | 55.24 | 33.71 | 53.65 |
| USC (Chen et al., 2024b) | 4 | 46.44 | 76.68 | 58.39 | 37.64 | 56.43 |
| GenRead (Yu et al., 2023a) | 2 | 45.08 | 77.39 | 54.90 | 34.27 | 54.70 |
| RobustRAG (Xiang et al., 2024)[7] | 11 | 34.24 | 67.49 | 44.06 | 32.02 | 45.59 |
| InstructRAG (Wei et al., 2024) | 1 | 46.78 | 80.57 | 54.90 | 34.83 | 56.14 |
| Self-Route (Xu et al., 2024) | 1-2 | 47.46 | 79.86 | 58.04 | 38.20 | 57.58 |
| ASTUTE RAG (t=1) | 2 | 50.17 | **81.63** | 58.04 | 40.45 | 59.21 |
| ASTUTE RAG (t=2) | 3 | **51.53** | 81.27 | 58.74 | 40.45 | **59.69** |
| ASTUTE RAG (t=3) | 4 | 48.47 | 80.21 | **60.14** | **42.13** | 59.21 |

Table 3: Main results on our Gemini under zero-shot setting.

**Performance on long-form QA.** We have conducted additional experiments on the long-form QA dataset, ASQA. The results in Table 2 demonstrate that ASTUTE RAG consistently achieves significant improvements in this new task, reinforcing its effectiveness across diverse scenarios.

**Worst-case performance on RGB.** Figure 4 presents the results under the worst-case setting on RGB where all retrieved documents are negative. It demonstrates the noise robustness of ASTUTE RAG and baseline RAG methods. The performance gap between RAG and No RAG exceeds 50 points, highlighting the detrimental impact of imperfect retrieval results and emphasizing the importance of providing robust safeguards against worst-case scenarios. While the baseline RAG methods outperform the original RAG, they still obviously fall behind No RAG. ASTUTE RAG is the only RAG method that reaches a performance close to No RAG under the worst-case scenario, further supporting its effectiveness in addressing imperfect retrieval augmentation.

### 4.3 ANALYSES

**Performance by retrieval precision.** We compare the performance of ASTUTE RAG and baselines across different subsets partitioned by their retrieval precision, on our collected data with Claude as the LLM. As shown in Figure 5, ASTUTE RAG achieves consistently better performance than all baselines across different retrieval precision, indicating its effectiveness in improving RAG trustworthiness in broad scenarios. Notably, ASTUTE RAG does not sacrifice performance gain under high retrieval quality in exchange for improvement under low retrieval quality. When the retrieval quality is extremely low (close to zero retrieval precision), all other RAG variants underperforms the 'No RAG' baseline, except for the proposed ASTUTE RAG. This observation aligns with the worst-case results on RGB. It demonstrates the difficulty in overcoming imperfect retrieval augmentation, and verify the effectiveness of ASTUTE RAG in doing so.

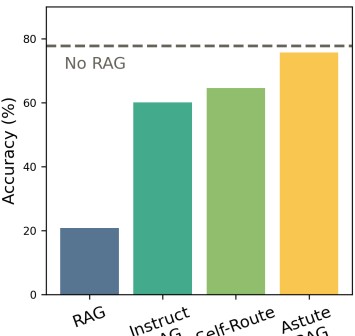

Figure 4: Worst-case performance of Claude on RGB. ASTUTE RAG reaches a performance close to No RAG, while other RAG systems are far behind.

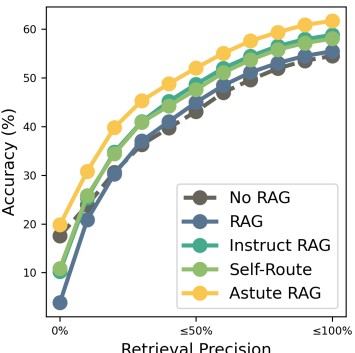

Figure 5: Performance across buckets by retrieval precision.

**Effectiveness in addressing knowledge conflicts.** We split our collected data in to three subset according to the answers from Claude, with and without RAG. The answers from two inference methods can be both correct, both incorrect, or conflicting with one being correct. These three subsets represents the three situations between internal and external knowledge. The results are shown in Figure 6. On the conflicting subset, ASTUTE RAG successfully chooses the correct answer

---

[7]We observe a high refusal rate in responses of RobustRAG.

in approximately 80% of cases, being the most effective method in addressing knowledge conflicts. Notably, ASTUTE RAG even brings performance improvement on the subset where neither internal nor external knowledge alone leads to the correct answer. This indicates that ASTUTE RAG can effectively combine partially-correct information from LLM-internal and external knowledge, to achieve the correct answer through collective information across them.

**Effectiveness of Adaptive Generation.** The results in Table 5 illustrate the model's performance when varying the maximum number of passages generated. The design of adaptive generation has been effectively reflected, as with the default setting ($\hat{m}$=1), the model is already able to dynamically change the number of generated passages.

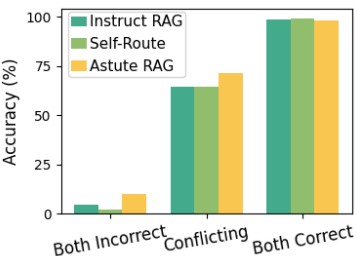

**Accuracy of Intermediate Steps.** To investigate the performance of intermediate steps, including knowledge consolidation and confidence assignment, we use LLM-as-a-judge with the instruction in Appendix A. Our experimental results show that the accuracy for knowledge consolidation is 98.2%, and for confidence assignment, it is 95.0%. These results demonstrate the effectiveness of the proposed framework in the intermediate stages. It also indicates that the current prediction errors are mainly due to the knowledge gaps instead of propagation of error from each step in our framework.

Figure 6: Performance on conflicting and consistent instances between No RAG and RAG.

**Efficiency by Tokens Consumed.** We present the average number of tokens used per instance in Table 6. Given that inference cost scales with the number of tokens, ASTUTE RAG incurs only a marginal cost increase of less than 5% while delivering a substantial relative improvement of over 11% compared to the RAG baseline.

**Influence of Passage Ordering Strategies.** We apply different ordering strategies introduced by Alessio et al. (2024), on RAG and ASTUTE RAG. As shown in Table 7, we find that the improvement with ASTUTE RAG is significantly larger than the gap between different ordering strategies. Moreover, the consolidation process makes ASTUTE RAG less sensitive to the order of passages.

**Qualitative study.** In Figure 7, we present two representative examples showing the intermediate outputs of ASTUTE RAG. In the first example, LLM without RAG generates a wrong answer, while RAG returns a correct answer. ASTUTE RAG successfully identified the incorrect information in its generated passage and an external passage, avoiding confirmation bias Tan et al. (2024). In the second example, LLM is correct but RAG is incorrect due to the noisy retrieval results. ASTUTE RAG detected the correct answer from noisy context by checking with its internal knowledge.

## 5 RELATED WORK

Retrieval augmented generation (RAG) seeks to address the inherent knowledge limitation of LLMs with passages retrieved from external sources of information such as private corpora or public knowledge bases (Guu et al., 2020; Lewis et al., 2020; Borgeaud et al., 2022). Given the widespread adoption of RAG in various real-world applications, including risk-sensitive domains, the negative impact of noisy information within retrieved passages has garnered increasing attention (Cuconasu et al., 2024). Recent work has sought to enhance the robustness of RAG systems against noise from various perspectives, including training LLMs with noisy context (Yu et al., 2023b; Yoran et al., 2024; Pan et al., 2024; Fang et al., 2024), training small models to filter out irrelevant passages (Wang et al., 2023b; Xu et al., 2023), passage reranking (Yu et al., 2024; Glass et al., 2022), dynamic and iterative retrieval (Jiang et al., 2023; Asai et al., 2023; Yan et al., 2024), query rewriting (Ma et al., 2023), and speculative drafting (Wang et al., 2024). These studies focus on distinct modules or stages of RAG systems and are orthogonal to our work.

Our work focuses on enhancing RAG robustness at the post-retrieval stage, after retrieved passages have been provided. On this topic, RobustRAG (Xiang et al., 2024) aggregates answers from each independent passage to provide certifiable robustness. InstructRAG (Wei et al., 2024) instructs the LLM to provide a rationale connecting the answer with information in passages. MADRA (Wang et al., 2023a) applies multi-agent debate to select helpful evidence. However, these works do not explicitly incorporate internal knowledge to recover from RAG failures and may therefore collapse

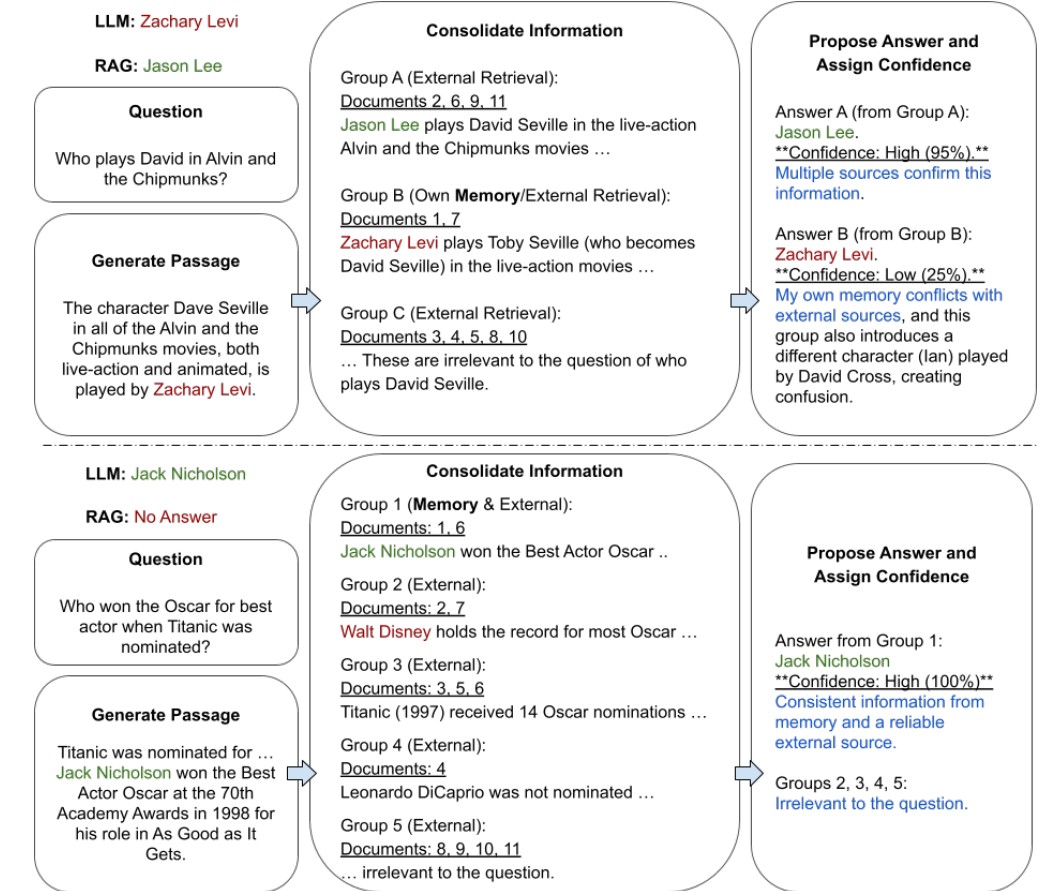

Figure 7: Qualitative examples. *Top:* ASTUTE RAG identified the error in internal knowledge (i.e., generated passage) by confirming with external sources. *Bottom:* ASTUTE RAG detected the correct answer from noisy retrieved information by checking with its internal knowledge. Standard RAG does not provide an answer because the retrieved passages are too noisy.

when the majority of retrieved passages are negative. In terms of emphasizing internal knowledge of LLMs in RAG, recent work has explored using LLM-generated passage as context (Yu et al., 2023a), training models to match generated and retrieved passages (Zhang et al., 2023), adaptively switching between LLMs with and without RAG (Xu et al., 2024; Mallen et al., 2023; Jeong et al., 2024), and combining answers from internal and external knowledge through contrastive decoding (Zhao et al., 2024; Jin et al., 2024). Different from prior work, we provide an in-depth analysis connecting imperfect retrieval, knowledge conflicts, and RAG failures. Specifically focusing on the imperfect context setting, our method is training-free and applicable to black-box LLMs, combines both internal and external knowledge, and offers broader usability and adaptability.

## 6 CONCLUSION

Our paper investigates the impact of imperfect retrieval on the performance of RAG systems and identifies knowledge conflicts as a key challenge. To address this, we introduce ASTUTE RAG, a novel approach that leverages the internal knowledge of LLMs and iteratively refines the generated responses by consolidating internal and external knowledge in a source way. Our empirical results demonstrate the effectiveness of ASTUTE RAG in mitigating the negative effects of imperfect retrieval and improving the robustness of RAG systems, particularly in challenging scenarios with unreliable external sources. Among the limitations, ASTUTE RAG's effectiveness hinges on the capabilities of advanced LLMs with strong instruction-following and reasoning abilities, hence potentially more limited applicability with less sophisticated LLMs. As an important future direction, extending the experimental setup to include longer outputs would be important, where the challenges of imperfect retrieval and knowledge conflicts may be even more pronounced.

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

## A PROMPT TEMPLATE FOR ASTUTE RAG

---

**Prompt for Adaptive Passage Generation ($p_{gen}$)**

Generate a document that provides accurate and relevant information to answer the given question. If the information is unclear or uncertain, explicitly state 'I don't know' to avoid any hallucinations.

Question: {question} Document:

---

**Prompt for Iterative Knowledge Consolidation ($p_{con}$)**

Task: Consolidate information from both your own memorized documents and externally retrieved documents in response to the given question.

* For documents that provide consistent information, cluster them together and summarize the key details into a single, concise document.
* For documents with conflicting information, separate them into distinct documents, ensuring each captures the unique perspective or data.
* Exclude any information irrelevant to the query.
For each new document created, clearly indicate:
* Whether the source was from memory or an external retrieval.
* The original document numbers for transparency.

Initial Context: {context}
Last Context: {context}
Question: {question}
New Context:

---

**Prompt for Knowledge Consolidation and Answer Finalization ($p_{ans}$)**

Task: Answer a given question using the consolidated information from both your own memorized documents and externally retrieved documents.

Step 1: Consolidate information
* For documents that provide consistent information, cluster them together and summarize the key details into a single, concise document.
* For documents with conflicting information, separate them into distinct documents, ensuring each captures the unique perspective or data.
* Exclude any information irrelevant to the query.
For each new document created, clearly indicate:
* Whether the source was from memory or an external retrieval.
* The original document numbers for transparency.

Step 2: Propose Answers and Assign Confidence
For each group of documents, propose a possible answer and assign a confidence score based on the credibility and agreement of the information.

Step 3: Select the Final Answer
After evaluating all groups, select the most accurate and well-supported answer.
Highlight your exact answer within <ANSWER> your answer </ANSWER>.

Initial Context: {context_init}
[Consolidated Context: {context}] # optional
Question: {question}
Answer:

---

---

**Prompt for Intermediate Step Evaluation**

**\*\*Task:\*\*** You are provided with the following:
1. A question.
2. The correct answer.
3. The input context.
4. The model's response, which contains:
- Consolidated context.
- Confidence scores for candidate answers.
Your task is to:
- Evaluate the \*\*quality of the consolidated context\*\* in the model's response and provide a label: '<consolidation> correct </consolidation>' or '<consolidation> incorrect </consolidation>'. This evaluation is only about whether the consolidation is correct given the input context.
- Evaluate the \*\*accuracy of the confidence score\*\* (whether it aligns with the confidence of the supporting context) and provide a label: '<confidence> correct </confidence>' or '<confidence> incorrect </confidence>'. The evaluation is only based on the consolidated context.
Note that correct consolidation and confidence do not necessarily indicate the correct answer.
Question:
{query}
Correct Answer:
{answer}
Input Context:
{input}
Model Response:
{response}
Evaluation:

---

## B DATA COLLECTION

Encompassing a *diverse* range of *natural* questions, our benchmark consists of *realistic* retrieval results with Google Search[8] as the retriever and the Web as the corpus. Notably, we do not select questions or annotate answers based on the retrieval results. This setting allows us to analyze the severity of imperfect retrieval in real-world RAG. It distinguishes our benchmark from previous ones that employ synthetic retrieval corruptions or that unintentionally reduce the frequency of imperfect retrieval with biased construction protocols (Chen et al., 2024a; Yang et al., 2024). Overall, our benchmark contains 1,042 short-form question-answer pairs, each paired with 10 retrieved passages.

**Question-answer pairs.** We consider question-answer pairs from four datasets of different properties spanning across general questions, domain-specific questions, and long-tail questions. NQ (Kwiatkowski et al., 2019) and TriviaQA (Joshi et al., 2017) are two widely-studied question-answering (QA) datasets in general domains. BioASQ (Tsatsaronis et al., 2015) is from biomedical domain that has demonstrated significant benefits from RAG when general-purpose LLMs are considered. PopQA (Mallen et al., 2023) focuses on long-tail knowledge and has been shown to be challenging for even advanced LLMs to solve without external knowledge. All these datasets contain questions with short-form answers and most of them list all valid answer variants. This format can support automatic verification of answer appearance in retrieved passages and model responses, leading to more precise evaluations.

**Retrieval process.** For each question in our benchmark, we query Google Search to retrieve the top 30 results and select the first 10 accessible websites. From each retrieved website, we extract the paragraph corresponding to the snippet provided in Google Search results as the retrieved passage. We do not consider enhancements to the retrieval side, such as query rewriting, as such enhancements are typically already incorporated into commercial information retrieval systems.

---

[8]https://developers.google.com/custom-search/v1/overview

## C  PERFORMANCE OF MISTRAL

| | NQ | TriviaQA | BioASQ | PopQA | Overall |
|---|---|---|---|---|---|
| *Mistral-Large (128B; version 2407)* | | | | | |
| RAG | 43.05 | 77.39 | 55.94 | 35.96 | 54.70 |
| Instruct RAG | 45.42 | 80.57 | 57.34 | 36.52 | 56.71 |
| Self-Route | 45.42 | 77.74 | 57.34 | 38.20 | 56.24 |
| Astute RAG | 50.17 | 82.69 | 58.39 | 42.13 | 59.88 |
| *Mistral-Nemo (12B; version 2407)* | | | | | |
| RAG | 39.32 | 66.78 | 48.95 | 32.58 | 48.27 |
| Instruct RAG | 38.31 | 61.84 | 50.35 | 23.60 | 45.49 |
| Self-Route | 41.36 | 73.50 | 51.75 | 30.90 | 51.15 |
| Astute RAG | 42.71 | 73.85 | 49.30 | 32.58 | 51.25 |

Table 4: Performance of Mistral-Large and Mistral-Nemo.

## D  EFFECTIVENESS OF ADAPTIVE GENERATION

| | NQ | TriviaQA | BioASQ | PopQA | Overall | #passages/instance |
|---|---|---|---|---|---|---|
| $\hat{m}=1$ | 52.20 | 84.10 | 60.14 | 44.38 | 61.71 | 0.69 |
| $\hat{m}=2$ | 52.20 | 85.16 | 60.84 | 43.26 | 62.00 | 1.24 |

Table 5: Performance and averge number of generaed passages using different $\hat{m}$.

## E  EFFICIENCY BY TOKENS CONSUMED

| | Overall Score | Avg Tokens |
|---|---|---|
| RAG | 55.47 | 1771 |
| Instruct RAG | 58.83 | 1953 |
| Self-Route | 58.06 | 1565 |
| Astute RAG | 61.71 | 1820 |

Table 6: Number of tokens used.

## F  PERFORMANCE BY ORDERING STRATEGIES

| Method | Ordering Strategy | NQ | TriviaQA | BioASQ | PopQA | Overall |
|--------|-------------------|----|----------|--------|-------|---------|
| RAG | Random | 43.39 | 76.33 | 56.99 | 34.83 | 54.61 |
| | Ascending | 43.05 | 75.62 | 57.69 | 34.83 | 54.51 |
| | Descending | 44.41 | 76.68 | 58.04 | 35.96 | 55.47 |
| | Ping-pong Descending Top-to-bottom | 44.75 | 77.39 | 57.69 | 35.96 | 55.66 |
| | Ping-pong Descending Bottom-to-top | 44.41 | 75.62 | 58.04 | 35.96 | 55.18 |
| AstuteRAG | Random | 51.86 | 84.81 | 61.19 | 41.57 | 61.61 |
| | Ascending | 51.86 | 85.51 | 59.79 | 42.13 | 61.52 |
| | Descending | 52.20 | 84.10 | 60.14 | 44.38 | 61.71 |
| | Ping-pong Descending Top-to-bottom | 52.20 | 84.45 | 59.09 | 43.82 | 61.42 |
| | Ping-pong Descending Bottom-to-top | 51.19 | 85.16 | 61.54 | 43.82 | 62.00 |

Table 7: Performance by Ordering Strategies.

