# OpenReview forum: "Astute RAG: Overcoming  Imperfect Retrieval Augmentation and Knowledge Conflicts for Large Language Models"
_ICLR.cc/2025/Conference — ICLR 2025 Conference Withdrawn Submission_

### Official Review · Reviewer_VS5X · 2024-10-28

**Soundness:** 3
**Presentation:** 2
**Contribution:** 2
**Rating:** 5
**Confidence:** 3

**Summary:**

In this paper, the authors address the imperfect retrieval problem and the knowledge conflict problem in RAG, and propose a AstuteRAG method which firstly adaptively generate passages using its internal parameter knowledge, then consolidate information from both generated passages and retrieved passages by taking their source information into consideration, constructing consistent groups, and addressing the conflicts iteratively. Experimental results show some performance improvements over several RAG baselines.

**Strengths:**

In general, I think it is critical to resolve  the imperfect retrieval problem and the knowledge conflict problem for better/more robust RAG, and the proposed pipleline(adaptive inner knowledge generation and iterative consolidation) is helpful  for robust RAG. Furthermore, I found the proposed method is training-free.

**Weaknesses:**

1. Both the imperfect retrieval problem and  the knowledge conflict problem have been widely recognized in the RAG field (see Benchmarking large language models in retrieval-augmented generation AAAI 24 for example), therefore it is inappropriate for "previous studies have rarely connected...". The authors should make a more comprehensive review of RAG studies;
2. There are many previous studies address the source/confidence-awareness of RAG,  iterative RAG, and knowledge-conflict consolidation methods. Therefore, to claim this paper's contributions, the authors should explain the differences/advantages of their methods over these studies.
3. Based on the above observations, I found the baselines in this paper are all general RAG baselines rather than RAG methods addressing the noise/conflict RAG problem,  which makes the experimental results not convincing enough. I think the authors should also compare with iterative-based, agent-based, and confidence-aware RAG baselines.

**Questions:**

See Weaknesses above.

---

> ### Author Response · Authors · 2024-11-25
>
> > **W1. Benchmarking large language models in retrieval-augmented generation. AAAI 24**
>
> We have acknowledged the contribution of previous work, including the AAAI '24 paper (lines 40 and 303). In fact, **we have evaluated our method on the RGB benchmark introduced in that paper** (see Figure 4). We find that, when the noise ratio is 1.0 (the worst-case where all passages are negative), all baseline RAG methods fall behind No RAG. Our method, Astute RAG, is the only RAG method that reaches a performance close to No RAG under the worst-case scenario, further supporting its effectiveness in addressing imperfect retrieval augmentation.
>
> While prior works have identified the issues of retrieval and knowledge conflicts, we are the first to provide a comprehensive and quantitative analysis connecting imperfect retrieval, knowledge conflicts, and RAG failures in real-world scenarios. In contrast, RGB relies on a synthetic noise ratio. Notably, the AAAI '24 paper does not propose solutions to these problems.
>
> > **W2. There are many previous studies. the authors should explain the differences/advantages of their methods over these studies.**
>
> We would be glad to include any overlooked related work if the reviewer could provide specific references.
>
> Currently, we have discussed relevant prior studies in the related work section. The key differences are:
>
> * We provide a **comprehensive analysis connecting** imperfect retrieval, knowledge conflicts, and RAG failures, going beyond previous analysis on isolated problems.
> * Building upon the insights from the analyses, our method encompasses a broader framework to address these challenges comprehensively under a **training-free and black-box setting**
> * Our method effectively **consolidates knowledge** from both internal and external sources and achieves more reliable answers with little additional cost.
>
> > **W3. the baselines in this paper are all general RAG baselines**
>
> The baselines in this paper include those specifically designed to address noise and conflict issues in RAG, such as RobustRAG and InstructRAG. Additionally, Self-Route incorporates confidence awareness. These recent baselines cover the types specified by the reviewer. Below is a brief description of them:
>
> * **RobustRAG (Xiang et al., 2024)** aggregates answers from each independent passage, using the majority consensus to prevent the negative influence of **malicious** passages.
> * **InstructRAG (Wei et al., 2024)** instructs the LLM to provide a rationale to identify **misleading or even erroneous** information in the retrieved contents.
> * **Self-Route (Li et al., 2024)** adaptively switches between LLMs with and without RAG based on the **model confidence**.
>
> We are happy to add more baselines if the reviewer can specify which ones they have in mind.

---

> ### Comment · Area_Chair_dXGr · 2024-11-26
> **Reminder: Rebuttal Deadline for ICLR 2025**
>
> Dear Reviewer VS5X,
>
> As the rebuttal deadline approaches, please kindly check the papers' discussion threads and respond to the authors' rebuttals. If you haven't had a chance to respond yet, I’d greatly appreciate your input soon. Your insights are invaluable to the authors and the review process.
>
> Thank you for your effort and support!
>
> Best regards,
>
> Area chair

---

> ### Author Response · Authors · 2024-11-27
>
> > **W2. There are many previous studies. the authors should explain the differences/advantages of their methods over these studies.**
>
> We sincerely thank the reviewer for the insightful feedback on improving the literature review. Below, we provide a more detailed discussion of related work referenced in our paper to clarify our positioning and highlight distinctions from prior research. We would be glad to include any overlooked related work if the reviewer could provide specific references.
>
> **Overview of Related Work**
>
> The negative impact of noisy information within retrieved passages has been increasingly scrutinized. Several orthogonal research directions address this issue at the retrieval stage of RAG, as outlined below:
> - **Dynamic Retrieval**: Iterative retrieval (Asai et al., 2023; Yan et al., 2024) retrieves additional information as needed during the generation process.
> - **Passage Reranking**: Techniques like those by Glass et al. (2022) and Yu et al. (2024) enhance robustness by refining retrieval quality through reranking.
> - **Query Enhancement**: Query rewriting methods, e.g., Ma et al. (2023), focus on dynamically refining input queries to improve retrieval outcomes.
> - **Speculative Drafting**: Wang et al. (2024) propose generating intermediate outputs to guide and refine the retrieval results.
> - **Training Passage Filters**: Approaches such as Wang et al. (2023c) train smaller models to identify and filter out unhelpful passages.
>
> These efforts primarily focus on improving retrieval and preprocessing stages to optimize the quality of passages fed into LLMs. While these methods enhance RAG performance, they address different research questions and are complementary to our work.
>
> **Positioning of Our Work**
>
> We tackle challenges at the **post-retrieval stage**, operating under the assumption that retrieved passages are already preprocessed and provided. This stage is **subsequent to the methods listed above**, making our approach **complementary and distinct**.
>
> **Closely Related Work**
>
> Several recent approaches address the noisy retrieval result issues in this post-retrieval setting are compared as baselines:
> - **GenRead** (Yu et al., 2023): Enhances retrieved passages by incorporating LLM-generated passages.
> - **RobustRAG** (Xiang et al., 2024): Aggregates answers from independent passages to ensure robustness, though it does not resolve conflicts using internal knowledge.
> - **InstructRAG** (Wei et al., 2024): Focuses on generating rationales that identify misleading or incorrect passages.
> - **Self-Route** (Xu et al., 2024a): Dynamically switches between LLMs with and without RAG based on model confidence of retrieved passages.
>
>
> **Less Directly Comparable Work**
>
> Other works in related domains make additional assumptions or address distinct settings:
> - **Training LLMs with Noisy Contexts** (Fang et al., 2024; Pan et al., 2024): Trains LLMs to directly handle noisy inputs, focusing on improving model robustness via supervised training.
> - **Contrastive Decoding** (Jin et al., 2024; Zhao et al., 2024): Assumes a setup where model logits are accessible, which is not applicable to most black-box LLMs.
> - **Matching Generated and Retrieved Passages** (Zhang et al., 2023): Involves training models to match retrieval outputs with generated content.
>
> **Summary**
>
> Our work uniquely focuses on the **post-retrieval stage**, addressing challenges that arise after preprocessing retrieved passages. By situating our method as the next step in the RAG pipeline, we build on and complement existing approaches.
> Currently, we have discussed relevant prior studies in the related work section. The key differences are:
>
> * We provide a **comprehensive analysis connecting** imperfect retrieval, knowledge conflicts, and RAG failures, going beyond previous analysis on isolated problems.
> * Building upon the insights from the analyses, our method encompasses a broader framework to address these challenges comprehensively under a **training-free** and **black-box** setting
> * Our method effectively **consolidates knowledge** from both internal and external sources and achieves more reliable answers with little additional cost.
>
> We hope this expanded discussion adequately addresses the reviewer’s concerns about the literature review. Please let us know if further elaboration is needed.

---

> > ### Comment · Reviewer_VS5X · 2024-11-28
> >
> > Thanks for the detailed rebuttal. Although the authors address some of my concerns, but my main concern are still here that the authors should better locate their contribution in a broad context, and the contributions can be significantly improved by addressing all concerns and adding their rebuttal into the next version of their paper.

---

> ### Author Response · Authors · 2024-11-28
>
> Dear Reviewer VS5X,
>
> Thank you for your valuable feedback on our paper. We have incorporated your suggestions and made the following updates to the pdf:
>
> - **Rephrased Contributions in a Broader Context**: Revised lines 46–50, 520–526, and updated Figure 1 to better situate our contributions.  The "Closely Related Work" are introduced and compared with in lines 310-342, the "Overview of Related Work" are introduced in lines 472-483, and the "Less Directly Comparable Work" are discussed in lines 483-526. **Methodologically, our work provides the following novel insights/benefits:**
>   - **Adaptive Generation**: The internal knowledge of LLMs is not equally accessible to all queries. Leveraging adaptive generation improves overall performance.
>   - **Iterative Source-Aware Consolidation**: Explicitly and iteratively instructing the model to consolidate internal and external knowledge in a source-aware manner enhances results. This approach differs from prior work, which typically resolves knowledge conflicts through methods such as pairwise matching or pre-defined switching rules.
>   - **Train-Free and Efficient**: Our method is train-free, applicable to black-box models, and achieves an 11% relative performance improvement with only a 5% increase in token cost, demonstrating significantly higher efficiency compared to baselines.
> - **Comprehensive Analysis of Advantages**: Provided a detailed comparison of AstuteRAG with baselines, focusing on adaptive generation, efficiency, and robustness to ordering strategies (lines 422–462, Tables 5, 6, and 7).
> - **Expanded Verification of AstuteRAG's Effectiveness**: Tested its performance on more open-source models of varying sizes (Table 4).
> - **Demonstrated Generalizability**: Evaluated AstuteRAG on long-form QA tasks, with results summarized in Table 2 and lines 400–402.
>
> As the due for updating the PDF is approaching, we are eager to know if there are any remaining concerns or areas for improvement. We are eager to further enhance our comparisons with prior work. If the reviewer can provide additional references or suggest specific works to include, we would be happy to incorporate them. Your guidance will be invaluable as we finalize the revisions.

---

> > ### Author Response · Authors · 2024-12-04
> >
> > Dear Reviewer VS5X,
> >
> > As the discussion period is ending, we would like to thank you for volunteering your time to review our paper and engaging in discussion. We hope to have answered all your questions and addressed the rest of the concerns you had.
> >
> > Thanks!

---

### Official Review · Reviewer_4FQF · 2024-10-31

**Soundness:** 3
**Presentation:** 3
**Contribution:** 3
**Rating:** 6
**Confidence:** 4

**Summary:**

The paper focuses on the problem of imperfect retrieval in RAG for LLMs. It analyzes its prevalence and negative impacts like RAG failures and knowledge conflicts. To address this, ASTUTE RAG is proposed, which adaptively generates internal knowledge, iteratively consolidates knowledge with source-awareness, and finalizes answers based on reliability. Experiments show ASTUTE RAG outperforms baselines in various scenarios, effectively resolving knowledge conflicts.

**Strengths:**

1. The research question, "Is there an effective method to integrate internal and external knowledge for enhancing the reliability of RAG?", is at the cutting edge of the field and holds substantial practical significance for the RAG community.

2. The essence of this paper is to utilize LLM to conduct a self-reflection on the query (generating internal knowledge documents) first, and then perform a conflict detection on external issues, making full use of the LLM's own capabilities and effectively alleviating the conflict between internal and external support.

3. The method enhances the robustness in case of retrieval failure. Even when all retrieved information is noisy, it can effectively ensure the bottom-line performance of LLM.

**Weaknesses:**

1. The performance and efficiency of the method are relatively low. The AstuteRAG proposed by the authors is used in the retrieval (online) stage, where LLM is made to generate as much internal knowledge Q&A as possible. This process is rather time-consuming and consumes tokens. Subsequently, iterative knowledge verification is required, which still uses LLM for conflict detection and paragraph merging. Overall, it is very time-consuming and reduces the practicality of the method.

2. A  major  potential drawback is that the knowledge consolidation process is fully completed by LLM. Although the conflicting information is grouped according to its consistency and the conflicting information is separated and then LLM is made to propose answers and assign confidence scores for each group of documents. Essentially, it is still LLM making judgments based on its own capabilities, and the influence of inherent bias and hallucination cannot be eliminated. A very concerning question is whether the confidence score of LLM shows an obvious bias when there is a conflict between internal and external knowledge? The bias may not necessarily be specifically towards internal knowledge or external knowledge, but rather towards some tokens with a specific high distribution.

3. The authors' verification of the method in this paper is all based on benchmarks with definite factual answers, lacking analysis of some scenarios. For example, both the internal knowledge and the external retrieval content are actually correct, but they form a conflict due to the time range or other specified limiting conditions in the query. From the current method, it is not seen that AstuteRAG has good robustness in this regard. This is also strongly related to the judgment basis mentioned in Weakness 2

**Questions:**

1. The relationship between retrieval imperfection and query characteristics needs further investigation. Please consider conducting additional experiments to: (1) analyze how different query types affect retrieval performance, (2) evaluate whether the findings from short-form QA can be generalized to other formats,(e.g. Long-form QA) and (3) compare system performance with varying numbers of retrieved documents (e.g., 10, 20, 50) given that modern LLMs can handle larger context windows. These analyses would provide more comprehensive insights into the retrieval mechanism's limitations and potential improvements.

2. In the manuscript, the author mentioned that there were several cases where the LLM refused to provide answers during the experiments. Could you please clarify how these refusal cases were handled in the results presented in Table 1? It would be valuable if you could provide specific statistics on the frequency of LLM refusals, and describe mitigation strategies implemented to handle these refusal scenarios? Furthermore, it appears that these LLM refusal cases could significantly impact the overall Recall of the question-answering , yet there seems to be a lack of discussion regarding this crucial aspect. A more detailed analysis of how these refusals affect the system's would strengthen the evaluation section and provide a more comprehensive understanding of the system's real-world performance  and limitations.

3. Please elaborate on the potential role of reranking in ASTUTE RAG. Specifically, could you: (1) discuss why reranking was not incorporated in the current implementation, (2) analyze how reranking might help address the performance decrease observed in Table 1 when using RAG, and (3) explore whether reranking could be a promising direction for future improvements in reducing noise and enhancing retrieval relevance. This discussion would provide valuable insights into the design choices of ASTUTE and potential avenues for enhancement.

4. Please maintain consistency in the abbreviation format, specifically for 'Retrieval-Augmented Generation (RAG)' in the first sentences of the abstract and  'Retrieval augmented generation (RAG)' in the first sentences of the introduction.

---

> ### Author Response · Authors · 2024-11-25
>
> > **W1. This process is rather time-consuming and consumes tokens.**
>
> We present the average number of tokens used per instance for Claude on our benchmark. Given that inference cost scales with the number of tokens, Astute RAG incurs only a marginal cost increase of less than 5% while delivering a substantial relative improvement of over 11% compared to the RAG baseline.
>
>
> |              | Overall Score | Avg Tokens |
> | ------------ | ------------- | ---------- |
> | RAG          | 55.47         | 1771       |
> | Instruct RAG | 58.83         | 1953       |
> | Self-Route   | 58.06         | 1565       |
> | Astute RAG   | 61.71         | 1820       |
>
> In addition, the efficiency of Astute RAG can be controlled by adjusting the consolidation iteration (t) and the max number of generated passages (m). Notably, even when both t and m are set to 1, the improvement is already significant.
>
> > **W2. whether the confidence score of LLM shows an obvious bias when there is a conflict between internal and external knowledge?**
>
> While our method relies on the LLM to assign confidence scores, we aim to mitigate this issue by explicitly consolidating potentially conflicting internal and external knowledge. This step helps reduce the direct influence of bias. As demonstrated in Figure 7, our method successfully generates the correct answer when either side is correct in these cases. The quantitative results in Figure 6 further show that our method significantly improves model performance on conflicting sets, which require the model to balance both sides effectively.
>
> However, we acknowledge that inherent biases in LLMs cannot be entirely eliminated, as they are intrinsic to the model's training and capabilities. The confidence scores generated by the LLM could indeed reflect biases related to token distribution, a known challenge in generative models [1,2]. Future work can explore robust training techniques to address these biases and improve the reliability of confidence assignments.
>
> [1] Xie, Jian, et al. "Adaptive chameleon or stubborn sloth: Revealing the behavior of large language models in knowledge conflicts." arXiv preprint arXiv:2305.13300 (2023).
>
> [2] Jin, Zhuoran, et al. "Tug-of-war between knowledge: Exploring and resolving knowledge conflicts in retrieval-augmented language models." arXiv preprint arXiv:2402.14409 (2024).
>
>
> > **W3/Q1. lacking analysis of some scenarios. evaluate whether the findings can be generalized to other formats,(e.g. Long-form QA)**
>
> We have conducted additional experiments on the long-form QA dataset, ASQA, following the setting in [1,2]. The results demonstrate that Astute RAG consistently achieves significant improvements in this new task, reinforcing its effectiveness across diverse scenarios.
>
> |  Method | EM ↑ |
> | --------------- | ----- |
> | RAG             | 32.97 |
> | Instruct RAG    | 34.99 |
> | Self-Route      | 34.47 |
> | Astute RAG      | 36.81 |
>
> [1] Wei, Zhepei, Wei-Lin Chen, and Yu Meng. "InstructRAG: Instructing Retrieval-Augmented Generation with Explicit Denoising." arXiv preprint arXiv:2406.13629 (2024).
>
> [2] Asai, Akari, et al. "Self-rag: Learning to retrieve, generate, and critique through self-reflection." arXiv preprint arXiv:2310.11511 (2023)

---

> ### Author Response · Authors · 2024-11-25
>
> > **Q1. compare system performance with varying numbers of retrieved documents**
>
> We report below the results using 5 and 10 retrieved passages, respectively. Astute RAG demonstrates consistent performance across these configurations, showcasing its robustness and adaptability to varying context lengths.
>
> |           |      | NQ    | TriviaQA | BioASQ | PopQA | Overall |
> | --------- | ---- | ----- | -------- | ------ | ----- | ------- |
> | RAG       | k=5  | 45.76 | 72.79    | 54.9   | 35.96 | 53.93   |
> |           | k=10 | 44.41 | 76.68    | 58.04  | 35.96 | 55.47   |
> | AstuteRAG | k=5  | 50.51 | 84.45    | 60.49  | 44.94 | 61.52   |
> |           | k=10 | 52.20 | 84.10    | 60.14  | 44.38 | 61.71   |
>
> > **Q2.1. how these refusal cases were handled in the results presented in Table 1?**
>
> Since the questions are answerable, refusals are counted as incorrect answers. This setting aligns with real-world helpfulness evaluation, where failing to provide an answer is considered a failure to assist.
>
> > **Q2.2. provide specific statistics on the frequency of LLM refusals**
>
> To assess the refusal rate, we count the ratio of responses containing a clear "no" or "not" in the final answer. We observe that there is no direct correlation between model performance and refusal rate. A method with better overall performance may also be more effective at identifying when additional information is needed, while a weaker method could either refuse more often or make more incorrect guesses.
>
> |              | refusal rate (%) |
> | ------------ | ------------ |
> | RAG          | 7.2          |
> | Instruct RAG | 6.2          |
> | Self-Route   | 4.2          |
> | Astute RAG   | 12.7         |
>
>
> > **Q3. Please elaborate on the potential role of reranking**
>
> We apply different ordering strategies introduced by [1], on RAG and our AstuteRAG. We find that the improvement with AstuteRAG is significantly larger than the gap between different ordering strategies. Moreover, the consolidation process makes Astute RAG less sensitive to the order of passages.
>
> |      Method     |          Ordering Strategy                          | NQ    | TriviaQA | BioASQ | PopQA | Overall |
> | --------- | ---------------------------------- | ----- | -------- | ------ | ----- | ------- |
> | RAG       | Random                             | 43.39 | 76.33    | 56.99  | 34.83 | 54.61   |
> |           | Ascending                          | 43.05 | 75.62    | 57.69  | 34.83 | 54.51   |
> |           | Descending                         | 44.41 | 76.68    | 58.04  | 35.96 | 55.47   |
> |           | Ping-pong Descending Top-to-bottom | 44.75 | 77.39    | 57.69  | 35.96 | 55.66   |
> |           | Ping-pong Descending Bottom-to-top | 44.41 | 75.62    | 58.04  | 35.96 | 55.18   |
> | AstuteRAG | Random                             | 51.86 | 84.81    | 61.19  | 41.57 | 61.61   |
> |           | Ascending                          | 51.86 | 85.51    | 59.79  | 42.13 | 61.52   |
> |           | Descending                         | 52.20 | 84.10    | 60.14  | 44.38 | 61.71   |
> |           | Ping-pong Descending Top-to-bottom | 52.20 | 84.45    | 59.09  | 43.82 | 61.42   |
> |           | Ping-pong Descending Bottom-to-top | 51.19 | 85.16    | 61.54  | 43.82 | 62.00   |
>
> [1] Alessio et al. 2024 Improving RAG Systems via Sentence Clustering and Reordering

---

> > ### Comment · Reviewer_4FQF · 2024-11-26
> >
> > Thank you for the authors' clarification. I have decided to maintain my positive review score.

---

> > > ### Author Response · Authors · 2024-11-26
> > >
> > > Dear Reviewer 4FQF,
> > >
> > > Thank you for your positive review and thoughtful feedback. We are more than willing to conduct additional experiments if you believe they would help further strengthen the paper and contribute to a higher score. Please let us know if there are specific areas where you would like to see further analysis, and we would be happy to accommodate.
> > >
> > > Thanks!

---

> ### Author Response · Authors · 2024-11-28
>
> > **W2 (cont) the knowledge consolidation process is fully completed by LLM. whether the confidence score of LLM shows an obvious bias**
>
> To further investigate whether the intermediate steps, especially knowledge consolidation and confidence assignment, exhibits obvious bias, we have evaluated the intermediate stages of knowledge consolidation and confidence assignment using LLM-as-a-judge with the following instruction:
> ```
> **Task:** You are provided with the following:
> 1. A question.
> 2. The correct answer.
> 3. The input context.
> 4. The model's response, which contains:
>    - Consolidated context.
>    - Confidence scores for candidate answers.
>
> Your task is to:
> - Evaluate the **quality of the consolidated context** in the model's response and provide a label: `<consolidation> correct </consolidation>` or `<consolidation> incorrect </consolidation>`. This evaluation is only about whether the consolidation is correct given the input context.
> - Evaluate the **accuracy of the confidence score** (whether it aligns with the confidence of the supporting context) and provide a label: `<confidence> correct </confidence>` or `<confidence> incorrect </confidence>`. The evaluation is only based on the consolidated context.
>
> Note that correct consolidation and confidence do not necessarily indicate the correct answer.
>
> Question:
> {query}
>
> Correct Answer:
> {answer}
>
> Input Context:
> {input}
>
> Model Response:
> {response}
>
> Evaluation:
> ```
>
> Our experimental results show that the accuracy for knowledge consolidation is 98.2%, and for confidence assignment, it is 95.0%. These results demonstrate the effectiveness of the proposed framework in the intermediate stages. It also indicates that the current prediction errors are mainly due to the knowledge gaps instead of  propagation of error from each step in our framework.

---

> > ### Author Response · Authors · 2024-12-04
> >
> > Dear Reviewer 4FQF,
> >
> > As the discussion period is ending, we would like to thank you for volunteering your time and engaging in discussion. We appreciate your positive review of our paper and hope we have answered all your questions and addressed any concerns you had.
> >
> > Thanks!

---

### Official Review · Reviewer_axAR · 2024-11-01

**Soundness:** 2
**Presentation:** 3
**Contribution:** 2
**Rating:** 5
**Confidence:** 4

**Summary:**

This paper proposes astute RAG, a framework to address imperfect retrieval and knowledge conflict challenges in retrieval-augmented generation.
Specifically, astute RAG consists of 3 steps including adaptive internal knowledge generation, iterative knowledge consolidation and answer finalization.
Experiments demonstrate that astute RAG outperforms previous robustness-enhanced RAG methods.

**Strengths:**

- This paper provides an empirical experimental analysis of the impact of imperfect retrieval and knowledge conflict on the accuracy of RAG systems, thereby validating the significance of its motivation.
- This paper presents an intuitive approach that, through the careful design of a framework and prompts, can alleviate the issues posed by the aforementioned challenges, thereby enhancing the performance of RAG systems without training.

**Weaknesses:**

- **Applicability of the method**: The framework proposed by the authors involves a complex pipeline and prompt design, which necessitates that the target model possesses various capabilities. However, the authors do not provide sufficient experimental evidence regarding the performance of current models in these intermediate stages. Furthermore, the experiments were conducted solely on two close-sourced models with undisclosed details. The lack of relevant details prevents readers from assessing the applicability and limitations of the method. I strongly recommend that the authors include experiments using more open-source models and provide analyses of the intermediate processes.
- **Mismatch between method and experimental design**: In the initial step, the authors employ adaptive generation to extract the model's internal knowledge. However, in the experimental setting, the model is required to generate a maximum of only one passage, which means the design of adaptive generation is not effectively reflected in the experiments. Consequently, the comparison of the number of API calls also lacks significance. Furthermore, I believe that the authors should focus on comparing the total number of API tokens used rather than the number of calls.
- **Novelty of the method**: Although the authors have designed a new framework, similar ideas have appeared in some prior works (e..g, [1]). This factor limits the novelty of this paper.

[1] Merging Generated and Retrieved Knowledge for Open-Domain QA

**Questions:**

see weakness

---

> ### Author Response · Authors · 2024-11-25
>
> > **W1. I strongly recommend that the authors include experiments using more open-source models.**
>
> We further conducted experiments using two open-source LLMs of different scales. The results remain consistent across both models. Specifically, for the more capable LLM, AstuteRAG significantly outperforms the baselines. For the relatively smaller model, AstuteRAG still demonstrates superior performance compared to the baselines on most datasets.
>
> | Mistral-Large (128B) | NQ    | TriviaQA | BioASQ | PopQA | Overall |
> | -------------------- | ----- | -------- | ------ | ----- | ------- |
> | RAG                  | 43.05 | 77.39    | 55.94  | 35.96 | 54.70   |
> | Instruct RAG         | 45.42 | 80.57    | 57.34  | 36.52 | 56.71   |
> | Self-Route           | 45.42 | 77.74    | 57.34  | 38.20 | 56.24   |
> | Astute RAG           | 50.17 | 82.69    | 58.39  | 42.13 | 59.88   |
>
> | Mistral-Nemo (12B) | NQ    | TriviaQA | BioASQ | PopQA | Overall |
> | ------------------ | ----- | -------- | ------ | ----- | ------- |
> | RAG                | 39.32 | 66.78    | 48.95  | 32.58 | 48.27   |
> | Instruct RAG       | 38.31 | 61.84    | 50.35  | 23.60 | 45.49   |
> | Self-Route         | 41.36 | 73.50    | 51.75  | 30.90 | 51.15   |
> | Astute RAG         | 42.71 | 73.85    | 49.30  | 32.58 | 51.25   |
>
>
> > **W2.1. the design of adaptive generation is not effectively reflected in the experiments**
>
> The results below illustrate the model's performance when varying the maximum number of passages generated. The design of adaptive generation has been effectively reflected, as with the default setting (m=1), the model is already able to dynamically change the number of generated passages.
>
> |  | NQ    | TriviaQA | BioASQ | PopQA | Overall | average number of generated passages per instance |
> | --------- | ----- | -------- | ------ | ----- | ------- | ------------------------------------------------- |
> | m=1       | 52.20 | 84.10    | 60.14  | 44.38 | 61.71   | 0.69                                              |
> | m=2       | 52.20 | 85.16    | 60.84  | 43.26 | 62.00   | 1.24                                              |
>
>
> > **W2.2. the authors should focus on comparing the total number of API tokens used**
>
> We present the average number of tokens used per instance for Claude on our benchmark. Given that inference cost scales with the number of tokens, Astute RAG incurs only a marginal cost increase of less than 5% while delivering a substantial relative improvement of over 11% compared to the RAG baseline.
>
> |              | Overall Score | Avg Tokens |
> | ------------ | ------------- | ---------- |
> | RAG          | 55.47         | 1771       |
> | Instruct RAG | 58.83         | 1953       |
> | Self-Route   | 58.06         | 1565       |
> | Astute RAG   | 61.71         | 1820       |
>
>
> > **W3. similar ideas have appeared in some prior works (e..g, [1])**
>
> We have acknowledged the contribution of [1] in the related work section (line 484). The key differences are as follows:
> * We provide an **in-depth analysis** linking imperfect retrieval, knowledge conflicts, and RAG failures.
> * Our study specifically focuses on the **imperfect context setting**, where the majority of passages may be incorrect.
> * Our method is **training-free** and applicable to **black-box** LLMs, offering broader usability and adaptability.
>
> [1] Merging Generated and Retrieved Knowledge for Open-Domain QA

---

> > ### Comment · Reviewer_axAR · 2024-11-28
> >
> > Thank you very much for your detailed responses. After reading them carefully, I still believe this paper needs further revision before being accepted, so I have decided to keep the original score.

---

> ### Comment · Area_Chair_dXGr · 2024-11-26
> **Reminder: Rebuttal Deadline for ICLR 2025**
>
> Dear Reviewer axAR,
>
> As the rebuttal deadline approaches, please kindly check the papers' discussion threads and respond to the authors' rebuttals. If you haven't had a chance to respond yet, I’d greatly appreciate your input soon. Your insights are invaluable to the authors and the review process.
>
> Thank you for your effort and support!
>
> Best regards,
>
> Area chair

---

> ### Author Response · Authors · 2024-11-28
>
> > **W3. similar ideas have appeared in some prior works (e..g, [1])**
>
> In the previous response, we provided a comparison based on research focus. Here, we delve into the differences in terms of technical novelty.
>
> **Methodologically, our work provides the following novel insights/benefits:**
>   - **Adaptive Generation**: The internal knowledge of LLMs is not equally accessible to all queries. Leveraging adaptive generation improves overall performance. In contrast, [1] generates a fixed number of passages.
>   - **Iterative Source-Aware Consolidation**: Explicitly and iteratively instructing the model to consolidate internal and external knowledge in a source-aware manner enhances results. In contrast, [1] relies on a finetuned passage matcher and does not explicitly or iteratively refine the context.
>   - **Train-Free and Efficient**: Our method is train-free, applicable to black-box models, and achieves an 11% relative performance improvement with only a 5% increase in token cost, demonstrating significantly higher efficiency compared to baselines.  In contrast, [1] need training and is not applicable to black-box models.
>
> We have made this clear in the updated paper pdf (lines 520–526).

---

> ### Author Response · Authors · 2024-11-28
>
> > **W1 (cont) the authors do not provide sufficient experimental evidence regarding the performance of current models in these intermediate stages**
>
> To address your concern, we have evaluated the intermediate stages of knowledge consolidation and confidence assignment using LLM-as-a-judge with the following instruction:
> ```
> **Task:** You are provided with the following:
> 1. A question.
> 2. The correct answer.
> 3. The input context.
> 4. The model's response, which contains:
>    - Consolidated context.
>    - Confidence scores for candidate answers.
>
> Your task is to:
> - Evaluate the **quality of the consolidated context** in the model's response and provide a label: `<consolidation> correct </consolidation>` or `<consolidation> incorrect </consolidation>`. This evaluation is only about whether the consolidation is correct given the input context.
> - Evaluate the **accuracy of the confidence score** (whether it aligns with the confidence of the supporting context) and provide a label: `<confidence> correct </confidence>` or `<confidence> incorrect </confidence>`. The evaluation is only based on the consolidated context.
>
> Note that correct consolidation and confidence do not necessarily indicate the correct answer.
>
> Question:
> {query}
>
> Correct Answer:
> {answer}
>
> Input Context:
> {input}
>
> Model Response:
> {response}
>
> Evaluation:
> ```
>
> Our experimental results show that the accuracy for knowledge consolidation is 98.2%, and for confidence assignment, it is 95.0%. These results demonstrate the effectiveness of the proposed framework in the intermediate stages. It also indicates that the current prediction errors are mainly due to the knowledge gaps instead of  propagation of error from each step in our framework.

---

> ### Author Response · Authors · 2024-11-28
>
> Dear reviewer axAR,
>
> We have updated the manuscript to address all the key points raised, including:
>
> - **Inclusion of more open-source models:** Results on Mistral-Large (128B) and Mistral-Nemo (12B) are provided in Table 4.
> - **Analysis of intermediate processes:** Lines 443-451 present evaluations of the accuracy of intermediate processes.
> - **Effect of adaptive generation:** Lines 437-442 and Table 5 show a performance comparison with varying maximum numbers of generated passages.
> - **Comparison of API tokens:** Lines 452-455 and Table 6 demonstrate that AstuteRAG achieves an 11% relative performance improvement with less than a 5% increase in token cost.
> - **Novelty comparison with [1]:** Lines 520-526 introduce [1] and clearly discuss the differences.
>
> We believe that all the issues raised by the reviewer have been thoroughly addressed. If further revisions are required, **we would greatly appreciate it if the reviewer could specify the requested changes.**

---

> > ### Author Response · Authors · 2024-12-04
> >
> > Dear Reviewer axAR,
> >
> > As the discussion period is ending, we would like to thank you for volunteering your time to review our paper and engaging in discussion. We hope to have answered all your questions and addressed the rest of the concerns you had.
> >
> > Thanks!

---

### Official Review · Reviewer_TZiG · 2024-11-03

**Soundness:** 3
**Presentation:** 2
**Contribution:** 2
**Rating:** 5
**Confidence:** 4

**Summary:**

The paper introduces Astute RAG, a novel approach addressing imperfect retrieval and knowledge conflicts in RAG systems.
The authors first conduct comprehensive analyses demonstrating that imperfect retrieval is prevalent in real-world RAG applications and identify knowledge conflicts between LLMs' internal knowledge and external sources as a key bottleneck. Authors then proposes Astute RAG, which adaptively elicits internal knowledge from LLMs, performs source-aware knowledge consolidation, and finalizes answers based on information reliability. Besides that, the authors demonstrate that Astute RAG outperforms previous robustness-enhanced RAG methods (especially in worst-case scenarios where traditional RAG approaches fail).

**Strengths:**

-  The paper presents an advanced approach to RAG systems, with a well-structured methodology for combining internal LLM knowledge with external sources. The three-step process demonstrates careful consideration of the challenges in knowledge integration.
- The authors provide comprehensive experimental results using state-of-the-art LLMs (Gemini and Claude) and multiple datasets (NQ, TriviaQA, BioASQ, PopQA), offering valuable insights into the method's performance across different scenarios.
- The work addresses a critical challenge in RAG systems: "the prevalence of imperfect retrieval and knowledge conflicts", backed by solid empirical evidence showing that roughly 70% of retrieved passages don't directly contain true answers.
- The authors' analysis using realistic conditions with Google Search as a retriever provides valuable insights for the research community, particularly in understanding real-world RAG system behavior.

**Weaknesses:**

- While technically sound, the paper doesn't sufficiently articulate why Astute RAG is particularly advantageous for real-world applications compared to simpler approaches, such as different passage ordering strategies (e.g., chronological, relevance-based, or clustered arrangements), varying quality of knowledge sources (from high-authority to potentially unreliable sources), or different passage selection methods. In fact, authors state that passages are presented by reversed order in the set of experiments, although no further positional dependence has been explored. Recent RAG studies has shown significant differences between distinct arrangement strategies (e.g. Alessio et al. 2024 Improving RAG Systems via Sentence Clustering and Reordering).
- The focus on short-form QA tasks limits understanding of the method's broader applicability. Imho, retriever improvements do not always translate into proportional gains in final answers, particularly in open-domain questions. in specialized tasks, even small improvements in retrieval can significantly boost final answers.
- The method's reliance on advanced LLMs with strong instruction-following capabilities significantly limits its applicability. The paper would benefit from addressing how the approach could be adapted for more resource-constrained or smaller specialized language models. Could you discuss strategies for adapting Astute RAG to resource-constrained environments?

**Questions:**

While the paper presents valuable contributions and strong empirical results, it falls somewhat short of the technical depth and theoretical foundations:
- Could you elaborate on your choice and implementation of Google Search as the retrieval method: Were there specific criteria for determining which snippets were included in the 10 selected passages? What were the specific reasons for choosing Google Search over other commercial search engines (e.g., Bing)? How did you handle sponsored or advertised content in the search results?
- How might the results differ with other retrieval approaches? How sensitive is the method to different prompt templates? What is the impact of different parameter settings (e.g., number of iterations, maximum generated passages) on performance and stability?
- How would Astute RAG perform on more complex tasks beyond short-form QA? Could you provide examples of failure cases when the system might not be appropriate to use?

---

> ### Author Response · Authors · 2024-11-25
>
> > **W1. Recent RAG studies has shown significant differences between distinct arrangement strategies (e.g. Alessio et al. 2024)**
>
> We want to clarify that these research directions are **orthogonal to the focus of our work**. RAG systems consist of multiple components, and users can always integrate complementary techniques (e.g., inference algorithms and ordering strategies) to further enhance performance. As stated in line 79, our work specifically focuses on combining internal and external knowledge during LLM inference, assuming the external context is already provided. We adopt the current default settings because **descending order has been identified as the best practice by prior work, including Alessio et al., 2024**.
>
> Nevertheless, we agree with the reviewer that additional experiments on more ordering strategies will provide further insights. Per the reviewer’s request, **we evaluate different ordering strategies introduced by Alessio et al., 2024**, on RAG and our AstuteRAG. We find that the improvement with AstuteRAG is significantly larger than the gap between different ordering strategies.
>
> |      Method     |          Ordering Strategy                          | NQ    | TriviaQA | BioASQ | PopQA | Overall |
> | --------- | ---------------------------------- | ----- | -------- | ------ | ----- | ------- |
> | RAG       | Random                             | 43.39 | 76.33    | 56.99  | 34.83 | 54.61   |
> |           | Ascending                          | 43.05 | 75.62    | 57.69  | 34.83 | 54.51   |
> |           | Descending                         | 44.41 | 76.68    | 58.04  | 35.96 | 55.47   |
> |           | Ping-pong Descending Top-to-bottom | 44.75 | 77.39    | 57.69  | 35.96 | 55.66   |
> |           | Ping-pong Descending Bottom-to-top | 44.41 | 75.62    | 58.04  | 35.96 | 55.18   |
> | AstuteRAG | Random                             | 51.86 | 84.81    | 61.19  | 41.57 | 61.61   |
> |           | Ascending                          | 51.86 | 85.51    | 59.79  | 42.13 | 61.52   |
> |           | Descending                         | 52.20 | 84.10    | 60.14  | 44.38 | 61.71   |
> |           | Ping-pong Descending Top-to-bottom | 52.20 | 84.45    | 59.09  | 43.82 | 61.42   |
> |           | Ping-pong Descending Bottom-to-top | 51.19 | 85.16    | 61.54  | 43.82 | 62.00   |
>
> > **W2/Q3. The focus on short-form QA tasks limits understanding of the method's broader applicability**
>
> We conduct additional experiments on the long-form QA dataset, ASQA, following the setting in [1,2]. The results demonstrate that Astute RAG consistently achieves significant improvements in this new task, reinforcing its effectiveness across diverse scenarios.
>
> |  Method | EM ↑ |
> | --------------- | ----- |
> | RAG             | 32.97 |
> | Instruct RAG    | 34.99 |
> | Self-Route      | 34.47 |
> | Astute RAG      | 36.81 |
>
> [1] Wei, Zhepei, Wei-Lin Chen, and Yu Meng. "InstructRAG: Instructing Retrieval-Augmented Generation with Explicit Denoising." arXiv preprint arXiv:2406.13629 (2024).
>
> [2] Asai, Akari, et al. "Self-rag: Learning to retrieve, generate, and critique through self-reflection." arXiv preprint arXiv:2310.11511 (2023).
>
> > **W3. Could you discuss strategies for adapting Astute RAG to resource-constrained environments?**
>
> We assume that the LLM possesses standard instruction-following capabilities, which is a common assumption when leveraging LLMs for QA tasks. It is unlikely that one would depend on answers from a weak model incapable of comprehending the question. To address concerns about the resource requirements of LLMs, we further conducted experiments using smaller, open-source LLMs, demonstrating the generalizability of Astute RAG to more resource-constrained environments.
>
> | Mistral-Large (128B) | NQ    | TriviaQA | BioASQ | PopQA | Overall |
> | -------------------- | ----- | -------- | ------ | ----- | ------- |
> | RAG                  | 43.05 | 77.39    | 55.94  | 35.96 | 54.70   |
> | Instruct RAG         | 45.42 | 80.57    | 57.34  | 36.52 | 56.71   |
> | Self-Route           | 45.42 | 77.74    | 57.34  | 38.20 | 56.24   |
> | Astute RAG           | 50.17 | 82.69    | 58.39  | 42.13 | 59.88   |
>
> | Mistral-Nemo (12B) | NQ    | TriviaQA | BioASQ | PopQA | Overall |
> | ------------------ | ----- | -------- | ------ | ----- | ------- |
> | RAG                | 39.32 | 66.78    | 48.95  | 32.58 | 48.27   |
> | Instruct RAG       | 38.31 | 61.84    | 50.35  | 23.60 | 45.49   |
> | Self-Route         | 41.36 | 73.50    | 51.75  | 30.90 | 51.15   |
> | Astute RAG         | 42.71 | 73.85    | 49.30  | 32.58 | 51.25   |

---

> ### Author Response · Authors · 2024-11-25
>
> > **Q1.1. Could you elaborate on your choice and implementation of Google Search as the retrieval method?**
>
> The details have been provided in the appendix B (lines 775-779). We will make them clearer in the updated version.
>
> >  **Q1.2. Were there specific criteria for determining which snippets were included in the 10 selected passages?**
>
> The snippets are returned by the Google Search API. We extract the paragraph including the snippet for each of top 10 retrieved results.
>
> > **Q1.3. What were the specific reasons for choosing Google Search over other commercial search engines (e.g., Bing)?**
>
> We chose Google Search as it is one of the most widely used search engine, making it an ideal testbed for our evaluation.
>
> > **Q1.4. How did you handle sponsored or advertised content in the search results?**
>
> The API does not return sponsored or advertised content [1].
>
> [1] https://support.google.com/adsense/thread/14297155/is-the-custom-search-api-response-contain-ads?hl=en
>
> > **Q2.1. How might the results differ with other retrieval approaches?**
>
> We emphasize that for quantitative analysis, the critical factor is not the choice of retriever itself but the retrieval quality it delivers. Retrieval quality, partially determined by the retrieval approach, directly impacts RAG performance. Analyzing performance based on retrieval quality provides a more rigorous evaluation. To demonstrate the method's generalizability across varying retrieval qualities, we report performance across different retrieval precision levels, as shown in Figure 5.
>
> > **Q2.2. How sensitive is the method to different prompt templates?**
>
> During development, we did not observe significant performance changes for slightly different expressions.
>
> > **Q2.3. What is the impact of different parameter settings (e.g., number of iterations, maximum generated passages) on performance and stability?**
>
> We have presented the results across varying numbers of iterations (t) in Tables 1 and 2. For reference, we include a subset of these results below.
>
> |  | NQ    | TriviaQA | BioASQ | PopQA | Overall |
> | --------- | ----- | -------- | ------ | ----- | ------- |
> | t=1       | 52.20 | 84.10    | 60.14  | 44.38 | 61.71   |
> | t=2       | 53.22 | 84.45    | 61.89  | 44.94 | 62.67   |
>
> We also conduct experiments to evaluate the impact of the maximum number of generated passages. The corresponding results are included below for reference.
>
> |  | NQ    | TriviaQA | BioASQ | PopQA | Overall | number of generated passages per instance |
> | --------- | ----- | -------- | ------ | ----- | ------- | ------------------------------------------------- |
> | m=1       | 52.20 | 84.10    | 60.14  | 44.38 | 61.71   | 0.69                                              |
> | m=2       | 52.20 | 85.16    | 60.84  | 43.26 | 62.00   | 1.24                                              |

---

> ### Comment · Area_Chair_dXGr · 2024-11-26
> **Reminder: Rebuttal Deadline for ICLR 2025**
>
> Dear Reviewer TZiG,
>
> As the rebuttal deadline approaches, please kindly check the papers' discussion threads and respond to the authors' rebuttals. If you haven't had a chance to respond yet, I’d greatly appreciate your input soon. Your insights are invaluable to the authors and the review process.
>
> Thank you for your effort and support!
>
> Best regards,
>
> Area chair

---

> > ### Author Response · Authors · 2024-12-03
> >
> > Dear Reviewer TZiG,
> >
> > Thank you for your time and thoughtful feedback on our paper. We hope our responses have addressed your concerns, and we kindly request that you consider updating the score accordingly.
> >
> > Thanks!

---

### Author Response · Authors · 2024-12-02
**Paper Update**

We have carefully addressed the reviewers' suggestions and incorporated the following updates into the revised PDF:

- **Open-Source Models**: Included results for Mistral-Large (128B) and Mistral Nemo (12B) in Table 4.
- **Long-form QA**: Added results on ASQA in Table 2 and updated lines 392–396.
- **Prior Work / Broad Context**: Enhanced discussions in lines 46–50 and 520–526, and updated Figure 1 to better contextualize our contributions.
- **Adaptive Generation**: Added detailed analysis in Table 5 and lines 427–442.
- **Intermediate Steps**: Provided additional insights in lines 443–450.
- **API Tokens**: Examined token consumption in Table 6 and lines 452–455.
- **Ordering Strategies**: Added results in Table 7 and lines 456–460.

These updates aim to address the reviewers' feedback comprehensively and strengthen the overall contribution of the paper.

---

> ### Author Response · Authors · 2024-12-04
> **Summary of Author Response**
>
> We have provided detailed responses to address the concerns and questions raised by each reviewer. Below is a concise summary of the key updates and clarifications:
>
> - **Reviewer [TZiG](https://openreview.net/forum?id=xy6B5Fh2v7&noteId=YOhj8eHKcw)**:  The main requests were to add results on different ordering strategies, long-form QA, and smaller LLMs. These have been addressed in **Table 7**, **Table 2**, and **Table 4**, respectively. Additional clarification questions have been thoroughly answered in a Q&A format.
>
> - **Reviewer [axAR](https://openreview.net/forum?id=xy6B5Fh2v7&noteId=YnAd9AgA8L)**:  The primary requests included adding open-source models, analyzing adaptive generation, examining intermediate steps, detailing the cost of API tokens, and comparing with prior work. We have addressed these in **Table 4**, **Table 5**, **lines 443–450**, **Table 6**, and **Section 5**.
>
> - **Reviewer [4FQF](https://openreview.net/forum?id=xy6B5Fh2v7&noteId=xmzuhr7b46)**:  The main requests focused on the cost of API tokens, analysis of intermediate steps, and long-form QA. These are now covered in **Table 6**, **lines 443–450**, and **Table 2**. Additional questions about the number of retrieved documents, refusal rates, and the role of passage ranking have also been clarified with supplementary results.
>
> - **Reviewer [VS5X](https://openreview.net/forum?id=xy6B5Fh2v7&noteId=tarNddm48p)**:  The primary concern regarding baselines and comparison with prior work has been thoroughly clarified in the responses.

---

### Note · Authors · 2024-12-16

I have read and agree with the venue's withdrawal policy on behalf of myself and my co-authors.